# Status and Magnitude of Grey Wolf Conflict with Pastoral Communities in the Foothills of the Hindu Kush Region of Pakistan

**DOI:** 10.3390/ani9100787

**Published:** 2019-10-11

**Authors:** Tauheed Ullah Khan, Xiaofeng Luan, Shahid Ahmad, Abdul Mannan, Waqif Khan, Abdul Aziz Khan, Barkat Ullah Khan, Emad Ud Din, Suman Bhattarai, Sher Shah, Sajjad Saeed, Ummay Amara

**Affiliations:** 1School of Nature Conservation, Beijing Forestry University, Beijing 100083, China; eco.tauheed@hotmail.com (T.U.K.); shahidbuneri87@hotmail.com (S.A.); 2Forest, Wildlife and Fisheries Department, Government of Punjab, Lahore 54500, Pakistan; mannan_shani@hotmail.com; 3Department of Zoology, Shaheed Benazir Bhutto University, Sheringal, Dir Upper 18000, Pakistan; wkhan6358@gmail.com; 4Laboratory of Animal and Human Physiology, Department of Animal Sciences, Quiad-i-Azam University, Islamabad 45320, Pakistan; abdalazeezkhan@gmail.com; 5Carnivores Conservation Lab, Department of Animal Sciences, Faculty of Biological Sciences, Quaid-I-Azam University, Islamabad 45320, Pakistan; barkat.paki@gmail.com; 6College of Environmental Science and Engineering, Beijing Forestry University, Beijing 100083, China; thegreatemi@yahoo.com; 7Institute of Forestry, Tribhuvan University, Kathmandu 44613, Nepal; sbhattarai@iofpc.edu.np; 8College of Forestry, Beijing Forestry University, Beijing 100083, China; sajjad_saeed222@yahoo.com (S.S.); sher378@gmail.com (S.S.); 9Department of Environmental Sciences, International Islamic University, Islamabad 45320, Pakistan, ummayamara@yahoo.com

**Keywords:** Human–wolf conflict, *Canis lupus*, livestock depredation, economic loss, Sheringal Valley

## Abstract

**Simple Summary:**

Despite higher loss due to disease, human–carnivore conflicts over livestock depredation is one of the major problems in carnivore conservation, both locally and globally. Locals share negative attitudes towards the wolf due to conflicts over livestock depredation. Using semi-structured questionnaires, we found that grey wolf is in a serious conflict with the locals, causing economic loss to them at the expanse of its own life. The locals considered the species a serious threat to their livestock, causing them economic losses, and wanted to reduce or even eliminate it from their area. Respondents having larger herd size and higher dependency on livestock for earning livelihoods shared more negative attitudes towards the wolves. In our study area the economic loss of the locals due to livestock mortalities from diseases was higher than that from wolf depredation. Therefore, we suggested that vaccination of the livestock and compensation schemes will help to change the perception of locals towards wolf.

**Abstract:**

Pastoralist–wolf conflict over livestock depredation is the main factor affecting conservation of grey wolf worldwide. Very limited research has been carried out to evaluate the pattern and nature of livestock depredation by wolf. This study aims to determine the status and nature of human–wolf conflict across different villages in the Hind Kush region of Pakistan during the period January 2016–December 2016. For this purpose, a total of 110 local male respondents from all walks of life were interviewed using a semi-structured questionnaire. The grey wolf was declared as a common species in the area by 51.3% of the locals with an annual sighting rate of 0.46 each. During the year (2016), a total of 358 livestock were lost to grey wolf predation and disease. Of the total livestock loss, grey wolf was held responsible for a total 101 livestock losses. Goat and sheep were the most vulnerable prey species as they accounted for 80 (79.2%) of the total reported depredations. Out of the total economic loss (USD 46,736, USD 424.87/household), grey wolf was accountable for USD 11,910 (USD 108.27 per household), while disease contributed 34,826 (USD 316.6 per household). High depredation was observed during the summer season 58.42% (n = 59) followed by spring and autumn. Unattended livestock were more prone to grey wolf attack during free grazing in forests. Most of the respondents (75.45%) showed aggressive and negative attitudes towards grey wolf. The herders shared more negative attitude (z = −3.21, *p* = 0.001) than businessman towards the species. Herders having larger herd size displayed more deleterious behavior towards wolves than those having smaller herd size. Active herding techniques, vaccinating livestock, educating locals about wildlife importance, and initiating compensating schemes for affected families could be helpful to decrease negative perceptions.

## 1. Introduction

Large carnivores are the top predators and considered the keystone species of an ecosystem. They keep an ecosystem in balance by regulating population of different species, mainly their prey [1,2,3]. Wolf species were once broadly distributed in most of the Nearctic and Palearctic biogeographic areas in the past [4,5]. Grey wolf belongs to the family Canidae, which is distributed throughout Pakistan [6]. Wolf is listed as endangered in the red list of Pakistan mammals [7], while it is rated as “Least Concern” globally [8]. It is one of the most controversial predators and is known for its predation on livestock and causing livestock owners serious economic losses [9].

The conversion of wolf habitat into agricultural land is the major cause of this decline in habitat range of wolf species. This ultimately induced and intensified human–wolf conflict over the livestock depredation and increased the dislike for the species among locals, which subsequently resulted in retaliatory wolf killings [10,11]. The conflict is also the main reason for the species extirpation across its western distribution range [12]. The species faced severe retaliatory killings all through its range using sanctioned firearms [13]. Closing or smoking out dens with adult wolf or pups and poisoning are some other ill-methods used to kill the species [12,14,15]. Beside retaliatory killings, there are some other threats that the species is facing, including habitat fragmentation and degradation, disease, decline of natural prey population, and competition with other carnivore species [16].

Human–wolf conflict is more intensified in the rural parts of different developing countries where locals are mostly dependent on their livestock for earning livelihoods [17]. There are mainly two underlying reasons: First, wolf predation on locals’ livestock causes a severe economic loss to the pastoral communities [18,19]. Secondly, wolves are infamous for occasional attacks on humans causing lethal injuries and even death in the worst cases [20,21], but these attacks are rare and often because of human interference, like destroying dens, traps, and persecution of pups [22,23]. Carnivores are responsible for about 3–18 % of annual economic loss to livestock-dependent families in the trans-Himalaya [24]. Moreover, surplus killings of livestock by wolf escalate the negative attitude among the herders. If in a single attempt a wolf kills more livestock than it needs, it causes a serious economic setback to the effected family [23,24,25]. The herders share a negative perception about the wolves because they mainly target the livestock. The higher the livestock density is, the higher the wolf predation, consequently leading to a negative attitude in the local people [26]. A rapid increase in human population and expansion of agricultural land into the grey wolf habitat has increased the chances of livestock depredation [27].

In Pakistan, the range of grey wolf is extended from the southern mountains of Baluchistan to the northern areas [6]. It inhabits the areas where locals are highly dependent on their agriculture activities and livestock rearing, where it is facing population due to conflict with locals with livestock depredation [22,24,28]. A camera trap and genetic based study conducted in its northern range in the country suggested a very thin population and a sporadic distribution [29,30] estimated that there are only 200 wolves in Pakistan, distributed across its whole range. A higher density (1.0–1.4/100 km^2^) of wolf has been reported in its northern range [31], while a relatively lower density (0.1/100 km^2^) was observed in its southern range [32]. The limited literature [33,34,35,36,37,38] on the subject in Pakistan describes that the wolf is the least accepted species among other sympathetic carnivores due to its conflict over livestock depredation with the locals. However, to our knowledge, studies regarding grey wolf conflict with humans in this region have not been reported. Therefore, this study aims to explore the status, magnitude, and nature of grey wolf conflict with pastoral communities in the Sheringal Valley.

## 2. Materials and Methods

### 2.1. Study Area

This study was carried out in the Sheringal Valley (35° 21.818’N, 72° 6.240’E) located in Dir Upper District, Khyber Pakhtunkhwa (KP) Province, Pakistan (Figure 1a,c,d). The valley spans over an area of 1972.7 km^2^ and has an elevation range of 1322–5444 m above sea level (Figure 1c). According to the recent census surveys carried out in 2017 by the Pakistan Bureau of Statistics, the total population of the area is 185,037 (density = 93/100 km^2^). It is surrounded by snowcapped or forest covered mountains. The northern part of the area is generally covered with forests. The long fluting River Panjkora runs from north to south bisecting the valley into two unequal parts. The human population resides on both sides of the river. Streams emerging from different watersheds join River Panjkora at respective points. The area has cool winters with temperature ranging from 11.22 °C–2.39 °C) and pleasant summers (29 °C) [38].

### 2.2. Field Survey

During December 2016, a total of 110 male participants (n = 110) were interviewed. We interviewed only male participants because in our study area, only males are engaged in outdoor activities related to livestock grazing and selling, fodder collection, and agriculture, while on the other hand females stay at home doing household jobs. The participants were orally interviewed informally and were selected based on their pre-existing knowledge about the presence of different wildlife species in general and grey wolf in particular. The main proportion of the participants included the herders, farmers, locals engaged in different businesses, school/college teachers, and local hunters of the study area. During the surveys, pre-designed questionnaires (Appendix A) having open ended questions were used to file responses of the respondents [26,36]. Questionnaire surveys are considered as an important tool to gather information about presence, tolerance, and perception of local communities towards the wildlife species present in an area [39]. Moreover, the local people can be a valuable and a reliable source of information about presence of wildlife species in their area [40,41]. To obtain a reliable dataset during the surveys, color photographs of the different wildlife species including grey wolf, snow leopard, common leopard, lynx, and red fox were shown to the respondents, who were asked to identify the grey wolf among the others. The respondents were also informed orally about the purpose of the study (i.e., the data will be used for scientific purpose and will not harm the community in future), to avoid exaggerated information. The questionnaire contained a total of 14 items including: the participant’s demographic data like age, house-hold size (HH), education level, profession, monthly income, number of earning members in family, agriculture land, numbers and types of livestock, and their dependency on the livestock they have; number of sighting of the grey wolf in the last year; status; perceptions about the grey wolf; and human attitudes towards the grey wolf. Intensity of wolf danger for livestock was categorized into five main categories: not dangerous, dangerous, slightly dangerous, very dangerous, and extremely dangerous [42]. All the study parameters were selected to explore the factors causing the conflict in this area because people’s attitudes are generally intricate; ethnicity, education, social factors, and different religious affiliations have been considered to shape the conflict intensity.

### 2.3. Data Analysis

The respondents’ attitudes toward grey wolves were categorized as neutral (i.e., respondent with unclear and wavering opinion), positive (i.e., shared positive views about wolfs), and negative (i.e., uncomplimentary opinion). For statistical analysis, we assigned scores of 0, 1, and −1 to neutral, positive, and negative attitudes, respectively. To identify the major factors that were influencing the respondents’ attitudes towards the grey wolf, we divided the variables into subcategories including respondent education level (educated = graduates, average educated = undergraduates, and uneducated = illiterate), age of a respondent (<30, 30–45, or >45 years), HH size (<5, 5–10, or >10), occupation (business, farmer, or employee), earning members (1, 3, or >3 members), agricultural land (1–2, 2–4, or >4 hectare), perception about wolf status (absent, rare, or common), sighting number of wolves (low; 0, medium; 1–3, or high; >3), number of livestock (>10, 10–30, or >30), and extent of livestock dependency (low = 1, medium = 2, or high = 3). Variables were selected by backward stepwise rejection method; this allowed us to identify important variables in a study. To examine the differences in respondents’ attitudes towards the grey wolf, Kruskal–Wallis ANOVA was applied. We also applied multiple post-hoc Wilcoxon rank tests to examine differences among different states of variables. Lastly, Pearson’s correlations were applied to find the correlating variables with attitude towards wolves. The statistical analyses were carried out using SPSS 20.0 (IBM Corp., Armonk, N.Y., USA). *p* < 0.05 denoted significant difference and data were offered as mean ± S.E. The map of the study area was developed using Arc Global Positioning System (ArcGIS, version 10.2, Environmental Systems Research Institute, Redlands, CA, USA).

## 3. Results

### 3.1. Demography of Local People

During the surveys a total of 110 male respondents were interviewed. The average age of the respondents was 38 years (SD = 14, range: 18–73 years). Amongst the respondents, most were herders and farmers (56.36%) followed by business men (25.45%), and 18.18% were college or university going students, school teachers, and other government servants. In our study area, 64.54% respondents showed high dependency on their livestock, 25.45% showed medium dependency, and 10.0% showed low dependency. Most of our respondents (71.10%) were knowledgeable (identification, presence, and status) about the wolves, whereas, 19.24% of locals were at average knowledge (identification, presence) level, and 9.66 % had poor knowledge (only identification).

### 3.2. Status of Livestock Holdings

The surveyed households (n = 110) owned a total of 4408 livestock (Table 1). Goats and sheep made the largest proportion of livestock 89.17% (n = 3931), followed by cattle 8.7% (384), and others (donkey, horse, and mule) 2.10% (93). The average herd size per household was 40.07 ± 7.51.

### 3.3. Livestock Depredation and Economic Loss

A total of 358 livestock losses were reported by respondents during the year 2016 to diseases and wolf predation. A total of 257 livestock losses were due to the diseases, while wolves were held responsible for 101 livestock depredations (Table 1). A total of 46,736 USD (424.87 USD/household) in economic loss was observed in a single year. Goats and sheep were the most vulnerable livestock to the grey wolf attacks, accounted for 80 (79.2%) of its killing, followed by cattle 12 (11.8%), and other 9 (8.9%).

### 3.4. Rate of Depredation across Seasons

Highest numbers of livestock depredations (58.42%; n = 59) by grey wolf were recorded in summer followed by winter 19.80% (20), spring 13.86% (14), and autumn 7.92% (8). Depredation was lowest during winter and spring (Table 2). Locals revealed that the majority (78.21%) of the livestock depredation incidents occurred inside the nearby forests/pastures where the livestock are set fee for grazing unattended and unguarded.

### 3.5. Wolf Sightings and Perceived Danger

A total of 51 wolf sighting records were reported by the respondents in one year with an average sighting rate of 0.46 per respondent. Our respondents shared mixed points of views about the status of wolves, as 51.3% declared that the species was commonly found in the area, while, 32.4% and 16.3 % of interviewees claimed that the species was rare or absent, respectively. All respondents declared that the intensity of wolf danger was very high for their livestock as compared to other carnivore species in the area. Occupation, perception towards wolf, and number of the livestock were the key parameters affecting the behavior (attitudes) of respondents towards the species (*p* = 0.001). Most of our respondents (75.45%) shared negative points of views about the wolves and they wanted to reduce or eliminate them from their area. There were a few respondents (14.55%) that remained neutral or opted not to express their attitude about the species, while a small fraction (10%) showed a positive attitude. To add to respondents’ views, those who thought that wolves were common in their area showed high levels of dislike compared to those who considered wolves absent (z = −6.33, *p* = 0.001). The farmers also had more negative attitudes than businessmen (z = −3.21, *p* = 0.001). Likewise, people that had greater number (i.e., *n* ≥ 30) (z = −1.73, *p* = 0.006) of livestock displayed more deleterious behavior towards wolves than those having lesser number of livestock (n = 0–10), (z = −3.73, *p* = 0.001), and (*n* = 10–30), (z = −2.73, *p* = 0.004). A strong correlation of livestock dependency (r = 0.551, *p* = 0.001), numbers of livestock (r = 0.635, *p* = 0.001), occupation of respondents (r = 0.575, *p* = 0.001), and number of earning members in family (r = −0.240, *p* = 0.023) was found with respondents’ attitudes towards grey wolves.

## 4. Discussion

The grey wolf population is being restored across developed countries due to better management and conservation. However, in countries like Pakistan, the species is still facing retaliatory killings due to livestock depredation [43]. The species has a declining population across its range in Pakistan. Our study area is within the global distribution range of the species [7] (Figure 1d). We carried out questionnaire surveys engaging the local people of the area in order to document the conflict, status, and attitude of the local community towards the species. According to [29] the northern parts of the country are the key and prime habitats of grey wolf. The presence of forests and pastures accompanied by rugged topography make the northern areas an ideal habitat for the species. However, these parts of the country also have heavily populated villages (Figure 1b,d) scattered within or very close to the grey wolf habitat.

Community-based surveys serve an opportunity for the locals to express their point of view and concerns about wildlife. The views of the locals cannot be neglected to design an effective wildlife management plan for an area [44,45]. In the study area, people showed a high dependency on livestock rearing. The estimated revenue generated from livestock rearing was USD 75,377, which translated into an average of 685.25 US$ per household annually (Table 1). In the year 2016 a total of 358 livestock losses were reported by the locals. Due to diseases, locals bear an economic loss of USD 34,826 (USD 316.6/household). The economic loss due to wolf depredation was USD 11,910 (USD 108.27 per household), which is almost three times lower than the loss due to diseases (Table 1). Previous literatures also supported the same results and acknowledged this notion [35,37,46,47,48]. The mountain communities already living below the poverty line [49,50,51] bear further economic stress due to diseases and livestock predators, which make the earning of monetary needs more difficult [46]. In this study, the respondents stated that a total of 101 livestock losses were caused due to wolf predation last year (Table 1). The increase in livestock and decline of natural prey species contributed to the escalated rate of depredation and human–wildlife conflict across many developing countries [52,53,54,55,56]. The situation accumulated in developing hatred towards the carnivores. Usually it is believed that the carnivores are the major and prime suspects of livestock losses and the paired economic losses. However, our results revealed that economic loss due to disease was higher than that due to wolf depredation. The area is lacking any livestock vaccination program or any veterinary services. The respondents reported that diseases that were apparently easily curable were not treated in time due to unavailability of the veterinary services.

Our results showed that the medium-sized livestock (goat and sheep) were relatively more prone to wolf attacks. Goats and sheep, due to their medium size body, weighing 25 kg (average), are easy to attack, capture, and kill as compared to large-sized livestock like cow, buffalo, horse, etc. [35,57]. Similar results were concluded in some other studies showing that goats and sheep fall within the preferred prey size range of carnivores [36,37,58].

There were noticeable differences in seasonal depredations, as the highest rate of depredation was recorded during the summer season, followed by autumn, spring, and winter, respectively. Local communities shift their livestock to nearby pastures and grazing grounds during summer and autumn. There in the pastures the livestock is usually left unattended as herders are engaged in other daily life activities. This situation turns the tables in favor of wolves and makes the unattended livestock more vulnerable to their attack. Studies carried out in different parts of the world reported a higher depredation rate when the livestock were left unguarded and unattended [59,60]. However, in winter they usually keep their livestock at home, feeding them with the stocked fodders gathered from the forests [35,61].

Studies carried out on human–wildlife conflicts concluded that wildlife-affected people hold negative feelings for carnivores [62,63,64,65]. In our study the interviewees shared a negative attitude towards the wolf and desired to reduce or eliminate it from their area. The unacceptability of the species among the locals is understandable because livestock rearing was the primary source of family revenue in the area. The wolf was declared as a significant predator in the area and locals considered wolves as a prime and lethal threat to their livestock. Additionally, the fear of wolves was higher than the damage they caused as they were portrayed as a sign of brutality in the study area. Despite the fact that not a single wolf attack on a human was reported during the surveys, locals still considered it dangerous for human lives. Peaceful coexistence between people and wolves is very challenging due to high rate of predation of livestock [66] by wolves; most livestock herders perceived wolves as dangerous to livestock and wanted to reduce or eliminate their population from the area. A recent study conducted in Karakoram suggested that livestock made about 66–75% of the diet of wolf and snow leopard [67], while in the Himalayas, livestock constituted about 24–27% of their diet [68]. Despite these challenges, conservationists are trying to help pastoralist communities through various incentive programs including predation compensation, rotational grazing, and community awareness programs [26]. Humans are usually tolerant towards the species where an economic return is involved. Vaccinating the livestock before or after the diseases also helps to change the perception of the locals about the predators. These efforts are helpful in altering human perceptions about carnivores [69]. Some wildlife conservation organizations, including the Snow Leopard Foundation, Panthera and Snow Leopard Trust, have carried out vaccination programs in some parts of the northern areas of Pakistan. Those programs helped to change the perception of local communities towards predators. The coexistence of predator and livestock can be attained by incorporating livestock management into conservation planning and initiation of predation mitigation and compensation schemes in the sensitive mountain ecosystem where pastoral communities live [70].

Our findings revealed that the families holding larger herd size and having a higher dependency on livestock rearing showed highly negative attitudes towards the grey wolf. In our study area most of the farmers (87%) that were illiterate and had high dependency on livestock showed negative attitudes. On the other hand, respondents who were reliant on professions and occupations other than herding showed acceptability for the species. Moreover, the educated respondents shared positive views about wildlife as compared to the illiterate respondents. People having larger herd size had an unwavering fear of wolf attacks and therefore they wanted to reduce the species from their area. Additionally, due to lack of any other alternative source of income they are highly dependent on the livestock to earn livelihoods [37,38]. The study also concluded that people with comparatively limited earning options, other than livestock rearing, expressed negative attitudes toward predators and vice versa. Literature suggests that formal and informal education helps to increase public understanding and acceptability of wildlife and is an effective solution to dilute people’s hatred for predators [18,37,38,70]. It increases the public understanding and tolerance of predators and plays a key role in equipping people with pro-conservation attitudes and practices. Education coupled with economic compensation for livestock losses and vaccination can help to gain public support for predator conservation. Initiating community learning sessions, engaging school and college going youth in conservation, and organizing other awareness raising events in the area will help to change the perception of locals towards grey wolf. 

## 5. Conclusions 

The findings of our study concluded that the local communities declared the grey wolf as a common carnivore species of the area. The locals bear a considerable amount of economic loss due to wolf depredation. Consequently, they hold severe negative attitudes towards the species and desired to reduce and eliminate it from their area. Based on our study and interactions with the local communities we recommend a few conservation measures to ensure the grey wolf conservation and minimize the economic losses of people. An effective livestock vaccination program is recommended to minimize livestock losses due to diseases. In cases of confirmed wolf depredation, the herds are recommended to be compensated for the loss. The compensation schemes are always very effective in changing the perception of locals about wildlife. Educating locals about the importance of wildlife and encouraging livestock guarding systems are also recommended to decrease risks of wolf attacks. Moreover, intensive sign and camera trap surveys are recommended to be carried out to determine the abundance and habitat preference of the species in the area.

## Figures and Tables

**Figure 1 animals-09-00787-f001:**
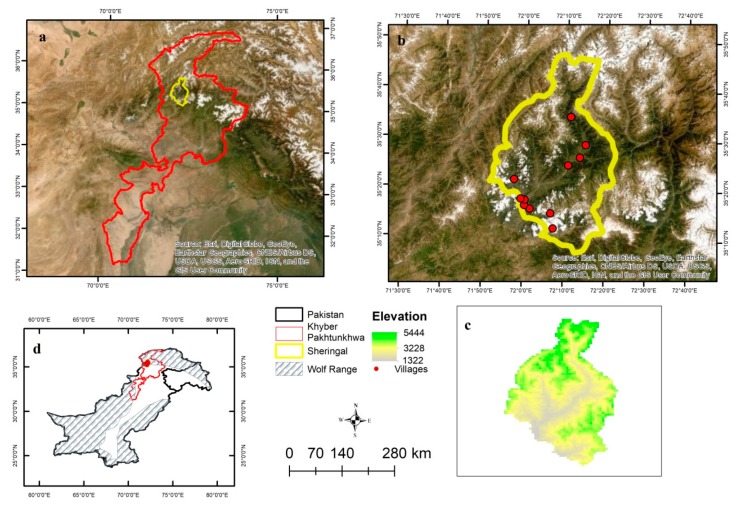
Location of our study area. Location of the study area (**a**), major villages of the study area (**b**), elevation range of the study area (**c**), and the distribution range of the grey wolf across Pakistan (**d**).

**Table 1 animals-09-00787-t001:** Detail of Livestock Holdings, Revenue Generated, and Economic Loss due to Disease and Grey Wolf per Household.

Livestock	No. of LS	LS Sold/Year	UV (US$)	Income (US$)	Income/HH (US$)	Disease		Wolf	
						No. of LS	Loss (US$)	No. of LS	Loss (US$)
Goat	1646	321	93	29,853	271.39	113	10,509	41	3813
Sheep	2285	387	103	39,861	362.37	101	10,403	39	4017
Cattle	384	17	331	5627	51.15	42	13,902	12	3972
Other	93	3	12	36	0.33	1	12	9	108
Total	4408	728		75,377	685.25	257	34,826	101	11,910
Total Economic Loss due to Disease and Wolf Depredation						46,736 (424.87/household)			

Abbreviations: UV: Unit value, US$ = United States Dollar, HH = households, LS = Livestock.

**Table 2 animals-09-00787-t002:** Rate of Grey Wolf Depredation in Different Seasons.

Season	Goat	Sheep	Cow	Other	Total Depredation	Percentage (%)
Summer	24	23	7	5	59	58.42
Winter	9	9	0	2	20	19.80
Spring	5	5	3	1	14	13.86
Autumn	3	2	2	1	8	7.92
Total	41	39	12	9	101	100

Other: Donkey, Horse etc.

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
