# Peer review of "Status and Magnitude of Grey Wolf Conflict with Pastoral Communities in the Foothills of the Hindu Kush Region of Pakistan"

_animals, 2019, doi:10.3390/ani9100787_

Round 1
Reviewer 1 Report
Review:
Status and Magnitude of Grey Wolf Conflict with Pastoral Communities in the Foothill of Hindu Kush Region of Pakistan
This article has merit in developing mitigation of human-wolf conflict in undeveloped areas and therefore warrants publication. However, I do have some comments regarding readability and I would like to see some details added to aid in understanding exactly how the study was run.
Firstly I would request the authors have the article proof read. For example in the simple summary alone I detected a number of English errors (see below). I will not comment on specifics regarding the English in the remainder of the article but in general it significantly affects the readability of the article.
Simple Summary: Despite higher loss due to disease, human-carnivore conflict over livestock depredation is one of the major problems in carnivore conservation both locally and globally. Locals share negative attitude towards the wolf due to conflict over livestock depredation. Using semi-structured questionnaires we found that the grey wolf is in a serious conflict with the locals, causing economical loss to them at the expense of its own life. The locals consider the species a serious threat to their livestock causing them economic losses and wanted to reduce or even eliminate it from their area. Respondents with larger herd sizes and higher dependency on livestock for earning livelihoods shared more negative attitude towards the wolves. We suggest that vaccination and compensation schemes will help to change the perception of locals towards the wolf.
Line 47: remove highlight
Line 62: I’m not sure what is meant here. Wolves signify destruction and negative change? How so? Change regarding what?
Line 76: As far as I’m aware, wolf attacks on humans are extremely rare globally (see Fritts et al, 2003 “Attacks by wild wolves are nonetheless rare, and fatal attacks are ever rarer and hard to document (note
especially Linnell et al. 2002 and McNay 2002a,b )”). The word notorious is too extreme here. Is this a particular problem in the study area? If so please do state that explicitly, with numbers. Otherwise, I would not consider attacks on humans as a major cause of wolf-human conflict, given how rare it actually is.
A little more detail in the introduction regarding wolves in Pakistan would be nice for those of us not familiar with the area. For example, how many wolves live there? Do they live close to human settlements? How does the wolf/human density compare to other geographical areas which experience similar human-wolf conflict?
Line 89/90: please cite the studies and give details as to what they studied and their findings.
Methods: is there any data on population sizes/density of wolves and humans in the study area? This would be helpful to know.
Please justify why only males were interviewed. What was the age range? What type of pre-existing knowledge did they have? Does this mean they had to know what a wolf was, or know about their ecology and movements? I assume this relates to lines 113-115, but this is not clear.
Even though the questions were open-ended and semi-structured, please state the questions asked in the interview. I don’t find lines 119-120 specific enough.
Figure 1 is very helpful. One small suggestion is to re-order so that (a) comes first, in the top left.
Lines 125-127: it would be helpful if some examples were given of what were considered neutral, positive or negative responses. Otherwise the categorisation seems arbitrary and subjective. I realise with semi-structured interviews a quantitative approach can’t be used, but more information on how the types of responses and how they were categorised would be useful.
Line 135: remove highlight
Line 145: remove brackets from 18%. In general, check the use of brackets, sometimes it’s incorrect.
Line 148-149: I still don’t understand what is considered knowledgeable.
Line 167: put (Table 2) in brackets.
Line 173-175: were the responses here related whether or not they had sighted a wolf? It could be expected for example that those who had seen a wolf were more likely to perceive high wolf presence.
Line 198-199: Agreed. This is the real strength of the study and could even be highlighted more in the introduction.
Line 204-205: It would be helpful to expand a little on how this study fits in with the bigger picture of the existing literature. Fitting it into the broader literature and comparing with other situations would inform on how best these results can be used for the most effective mitigation of human-wolf conflict.
I found the conclusion concise and effective.
Reviewer 2 Report
This study provides a potentially interesting insight into attitudes towards grey wolves in an area where people’s livelihoods are strongly dependent upon livestock. Whilst this study has potential to be of use in reducing livestock loss, improving attitudes to wolves and thereby improving conservation efforts for grey wolves in this region, the paper itself is lacking in important details which help the reader to understand how the authors implemented the study. These details are described below. I believe that this paper requires substantial revisions with especial attention to detail before it can be suitable for publication in Animals.
Simple summary.
Whilst the abstract makes mention of the issues of disease affecting livestock, the simple summary does not mention except to recommend vaccinating livestock to reduce loss. The issues of disease should be explained in a prior sentence to clarify why vaccination is necessary.
Introduction.
Line 76. ‘wolves are notorious for attacks on humans causing lethal injuries and even death in worst cases’. Some statistics to help define how frequently this happens would be helpful here.
Line 77: ‘Carnivores are responsible for about 3-18 % annual economic loss to the livestock depending families in the trans-Himalaya’ – but what about wolves specifically?
Line 89. ‘Up to date a few studies have been reported to assess the human-wolf conflict in Pakistan.’ It would be good to be see some description of these studies.
Line 102: 29 degrees C – is this the mean temp, or max? Need to specify.
Methods.
How were participants recruited?
I would like to see more description of the questionnaire, such as how many items it had, what these items assessed and some examples.
How is highly educated/average educated defined? How is extent of livestock dependency defined?
Line 136. Why was a Kruskal-wallis used? Could the authors please define what a backwards step-wiser rejection method is? Not all readers will be familiar with this. If multiple Wilcoxon rank tests were run, I would suggest to control for multiple tests to avoid inflating p values and producing false positives. What were the dependent and independent variables in these analyses? What variables affected positive attitudes? Do age or education, for example, play a role? (in the discussion, line 243, the effect of education is mentioned, but not supported by any data). I would also appreciate some figures to support these findings.
Figure 2 does not contribute anything that we can’t glean from table 2 and should be deleted. Instead I would suggest presenting figures of how different variables correlate or not with attitudes towards wolves.
Figure 1: what does KPK stand for?
Results.
SD should ideally be provided for age.
Line 148. How do you define knowledgeable?
What type of diseases effect livestock?
Line 177. ‘Occupation, perception towards wolf and number of the livestock were the key parameters effecting the behavior (attitude) of respondents towards the species’. This is very vague. How did occupation or perception affect attitudes? Why is only the p value reported and not additional statistics such as effect size?
Discussion.
Line 214. Given that disease is responsible for more livestock deaths than predation, is it possible that negative attitudes towards wolves could be reduced by providing locals with education on the numbers of losses due to disease versus predation? i.e. if they see that losses to wolves are much lower than losses to disease, is there a chance this could reduce targeting of wolves?
Line 247: what would this education involve, how would it be implemented?
In general I would like to see more detailed discussion of suggestions of how to reduce livestock loss and improve attitudes of locals towards wolves. How could the suggested changes be implemented? What would be the next steps of this research?
English.
There are a number of grammatical issues with the English as well as several misspelled words throughout, some of which I have detailed below. I advise the authors to have a native English speaker proof read the manuscript for them upon revising.
Line 29, 191. Questioners – should be questionnaires
Line 32. Earing – should be earning
Line 68. Sever – should be severe
Line 135. Extant – should be extent
